# Using DNA metabarcoding and direct behavioural observations to identify the diet of proboscis monkeys (*Nasalis larvatus*) in the Kinabatangan Floodplain, Sabah

**Valentine Thiry**[1,2]☺*, **Arthur F. Boom**[ID][3,4]☺*, **Danica J. Stark**[ID][5,6], **Olivier J. Hardy**[3], **Roseline C. Beudels-Jamar**[2], **Regine Vercauteren Drubbel**[1], **Sylvia Alsisto**[7], **Martine Vercauteren**[1], **Benoit Goossens**[5,6,7]*

**1** Anthropology and Human Genetics Unit, Faculté des Sciences, Université Libre de Bruxelles, Brussels, Belgium, **2** Conservation Biology Unit, O.D. Nature, Royal Belgian Institute of Natural Sciences, Brussels, Belgium, **3** Evolutionary Biology and Ecology Unit, Faculté des Sciences, Université Libre de Bruxelles, Brussels, Belgium, **4** Royal Museum for Central Africa, Biology Department, Section Vertebrates, Tervuren, Belgium, **5** Danau Girang Field Centre, c/o Sabah Wildlife Department, Kota Kinabalu, Malaysia, **6** Organisms and Environment Division, Cardiff School of Biosciences, Cardiff University, Cardiff, United Kingdom, **7** Sabah Wildlife Department, Kota Kinabalu, Malaysia

☺ These authors contributed equally to this work.
* valentinethiry89@gmail.com (VT); boomarthur@gmail.com (AFB); goossensbr@cardiff.ac.uk (BG)

## Abstract

Characterizing the feeding ecology of threatened species is essential to establish appropriate conservation strategies. We focused our study on the proboscis monkey (*Nasalis larvatus*), an endangered primate species which is endemic to the island of Borneo. Our survey was conducted in the Lower Kinabatangan Wildlife Sanctuary (LKWS), a riverine protected area that is surrounded by oil palm plantations. We aimed to determine the diet of multiple proboscis monkey groups by using two methods. First, we conducted boat-based direct observations (scan and *ad libitum* sampling) and identified 67 plant species consumed by the monkeys at their sleeping sites in early mornings and late afternoons. Secondly, we used the DNA metabarcoding approach, based on next-generation sequencing (NGS, MiSeq Illumina) of faecal samples (n = 155), using the short chloroplast sequence, the trnL (UAA) P6 loop. In addition, we built a DNA reference database with the local plants available in the LKWS. When combining feeding data from both methods, we reported a diverse dietary ecology in proboscis monkeys, with at least 89 consumed plant taxa, belonging to 76 genera and 45 families. Moreover, we were able to add 22 new genera as part of the diet of this endangered colobine primate in the LKWS. The two methods provided congruent and complementary results, both having their advantages and limitations. This study contributed to enhance the knowledge on the feeding ecology of proboscis monkeys, highlighting the significance of several plant species that should further be considered in habitat restoration plans or corridor establishment.

**Data Availability Statement:** DNA sequences generated for the LKWS database includes 205 plant species, belonging to 146 genera and 61 families. Data can be found on GenBank (accession numbers from OR670713 to OR670948)

**Funding:** The following funders had no role in study design, data collection and analysis, decision to publish, or preparation of the manuscript: FNRS (Fonds de la Recherche Scientifique); The Fonds Léopold III – pour l'Exploration et la Conservation de la Nature asbl; The FNRS Gustave Boël-Sofina Fellowship.

**Competing interests:** The authors have declared that no competing interests exist.

# Introduction

Accurate knowledge of the diet of endangered animal species is essential to develop appropriate conservation strategies [1, 2]. For herbivores, identifying which plant species are eaten is useful to target the plants that should be favoured in further conservation and management plans [3–5]. There are numerous direct and indirect, invasive and non-invasive, methods to study diet composition of wild animals [6]. The simplest way is the direct observation of feeding behaviour, though this method is not an easy task with elusive or nocturnal animals, or in difficult field conditions (e.g., inundated forest, many vines) [7]. In the latter cases, researchers should use a multidisciplinary approach, combining non-invasive indirect methods that provide complementary feeding data [6]. In fact, faecal analyses are good alternative approaches unveiling herbivorous diets [7], which can take diverse forms: (1) microhistological analyses examining plant cuticular fragments [8–10]; (2) alkane profiles of plant cuticular wax [11]; (3) near infrared reflectance spectroscopy (NIRS) [12, 13] and (4) DNA-based analyses coupled with either next-generation sequencing (NGS) targeting specific or universal primers (DNA metabarcoding) [7], or direct shotgun sequencing (metagenomics) [14]. Previous studies reported that DNA-based analyses provided diet data (list of ingested species) consistent with results obtained using direct behavioural observations [14, 15].

DNA metabarcoding refers to the identification of multiple species from a sample containing degraded DNA (e.g., faeces, soil, etc.); its emergence has been facilitated by the development of new sequencing technologies [16]. The method has already been used successfully to determine the diet of many herbivores [10, 17–22] At first, the chloroplast mini-barcode rbcL was initially used in studies using degraded DNA (i.e., coprolites or faeces) [23, 24], but it only allowed identification to the family level [25]. Since then, the most used DNA barcode in herbivory studies is the P6 loop of the chloroplast trnL (UAA) intron, a short DNA fragment [10–143 bp] designed by Taberlet and colleagues [25, see 21]. It exhibits several suitable features for metabarcoding studies, though its taxonomic resolution is limited. First, the sequences used for primers design (g and h primers) are extremely well conserved among spermatophytes, allowing for universal PCR amplification for a large number of plant taxa. Second, PCR amplifications are robust even for degraded DNA. Finally, trnL (UAA) intron sequences are among the most readily available chloroplast sequences in public databases, potentially allowing taxonomic identification until the genus or species level [25]. As an alternative to plastid DNA, the first and second internal transcribed spacers of nuclear ribosomal DNA (ITS1 and ITS2) were recently used to carry out diet analyses on the Italian hare (*Lepus corsicanus*), stock doves (*Columba oenas*), and Telfairs' skinks (*Leiolopisma telfairii*), providing unprecedented taxonomic resolution [21, 26]. However, to reach such high taxonomic resolution, a comprehensive reference DNA database containing the sequences of most plants available in a site is indispensable [7].

As a powerful non-invasive technique, the DNA metabarcoding can provide dietary information for large numbers of individuals simultaneously, contributing to increased knowledge on feeding ecology at a population scale [15]. However, it also presents some limitations. Indeed, the method is not appropriate for the quantification of herbivorous diets. It does not allow to identify which plant part has been consumed, and as the method is mostly based on chloroplast genes (i.e., rbcL, trnL), plant species from which chloroplast-rich items are consumed (e.g., leaves or stems) may be overrepresented, as compared to species mostly eaten for their fruits, seeds or roots [7, 24]. Moreover, it is usually not reliable to correlate proportions of DNA sequences retrieved from faeces to the proportions of items ingested by the animal (i.e., proportions of sequences differ from dietary mass proportions) [27].

The proboscis monkey (*Nasalis larvatus*) is an endangered colobine primate endemic to Borneo, where the species inhabits riverine, swamp and mangrove forests [28]. In riverine habitats, proboscis monkeys are highly associated with water bodies, foraging up to 800 m inland from the riverbank during the day and returning to the riverside where they spend the night [29], although inland sleeping occurs more as the moon waxes and temperatures are high [30]. Therefore, many previous studies were carried out by conducting boat-based surveys in early mornings and late afternoons [31–37]. However, a recent long-term study using GPS radio-telemetry technology on 10 collared individuals provided comprehensive data on proboscis monkey spatio-temporal habitat use in riverine forests: the mean daily distance travelled was 940 m (mean range: 285–2208 m) [30] and monkeys' home range averaged 81 ha [38].

Similar to other colobines, proboscis monkeys have a large sacculated forestomach wherein food fermentation occurs [39, 40], and therefore they usually avoid feeding on ripe fleshy fruits which are rich in sugar susceptible to rapid fermentation, thus producing gas, with deleterious effects on their digestive system [41]. Proboscis monkeys mostly consume young leaves, unripe fruits and seeds [42]. However, most knowledge on proboscis monkey diet is provided by short-term studies [31, 32] or is restricted to the riverside [33, 34, 43], except for the studies conducted by Boonratana [44] and Matsuda and colleagues [42] in Sabah. The latter is the most comprehensive study, describing the diverse and flexible diet of proboscis monkeys in riverine habitat (n = 188 consumed plant species), although this study was only based on seven adult proboscis monkeys belonging to one single habituated group [42].

In this paper, we aimed to report the diet composition of multiple groups of non-habituated proboscis monkeys, inhabiting the Lower Kinabatangan Wildlife Sanctuary, a protected area surrounded by oil palm plantations, located in Sabah, Malaysia, based on two methods: direct behavioural observations and DNA metabarcoding using trnL. Revealing the diet of the proboscis monkey using DNA metabarcoding is a considerable challenge, especially regarding the highly diverse diet of this colobine (i.e., 188 food plants [42]) and its long mean gut retention times (MRTs of 40 hours [45]), potentially resulting in high DNA degradation [14]. DNA metabarcoding has already been used to determine the diet of several colobine species: the Douc langur (*Pygathrix nemaeus*) [46], the banded leaf monkey (*Presbytis femoralis*) [14], and the black and white colobus monkey (*Colobus guereza*) [24], as well as of some folivorous and/ or frugivorous primates, such as golden-crowned sifakas (*Propithecus tattersalli*) [15], long-tailed macaques (*Macaca fascicularis*) [22], wild stump-tailed macaques (*M. arctoides*) [47], wild white-faced capuchins (*Cebus capucinus*) [48], Western gorillas (*Gorilla gorilla*) [24], and bonobos (*Pan paniscus*) [49].

In this study, we carried out numerous feeding observations of proboscis monkeys when refuging at their sleeping site along the Kinabatangan River, a region where the proboscis monkey population reaches nearly 2,000 individuals [50]. We also conducted a comprehensive sampling of proboscis monkey fresh faeces, over a wide study area, in parallel to building a local DNA reference database with the plants available in our study site. Firstly, we described the proboscis monkey diet data obtained by each of the two methods, listing the plant species consumed. As proboscis monkey diet is reported to change seasonally, in relation to food availability [34, 42], we looked for diet variation between seasons. Then, we compared the consistency of the plant taxa detected by both methods. Based on previous diet studies comparing DNA metabarcoding method to traditional techniques (field observations or microhistology) [10, 14], we would expect to record more plant taxa by DNA metabarcoding than by direct behavioural observations, especially in regard to the restricted riverine location of our direct observations in comparison to DNA-based analyses using faecal samples, potentially covering whole day feeding events (i.e., riverside and inland forest habitats). However, taking the low discriminant power of the trnL P6 loop into account, we would expect the taxonomic

resolution to be lower using the DNA metabarcoding approach than the direct behavioural observations. Finally, we combined feeding data obtained by both methods and discussed our approach by comparing our results to previous studies describing proboscis monkey diet in a proximate study site (Sukau, in [42, 44]).

## Methods

### Study site

This study took place over 15 months (May to August 2015, January to June 2016 and November 2016 to March 2017) in the Lower Kinabatangan Wildlife Sanctuary (LKWS, 5˚10′–05˚50′N, 117˚40′–118˚30′E), in Sabah (Malaysian Borneo) (Fig 1). Daily rainfall, minimum and maximum temperatures were measured at the research station. Mean (± SD) monthly rainfall averaged 176 ± 118 mm. Below, we refer to the wet season (November to March), where mean monthly rainfall reached 228 ± 134 mm, and the dry season (April to August), where it reached 117 ± 62 mm. Mean minimum and maximum temperatures averaged 24.5 ± 0.7 and 30.4 ± 1.8˚C, respectively.

### Vegetation survey

We conducted a vegetation survey in Lot 6 of the LKWS, using plot-sampling method. Using QGIS, we first drew parallel transect lines (west-east oriented), spaced by 750 m. We then set up 21 plots (50 m x 50 m) spaced by 750 m, along the transect lines. Four additional plots were finally set up between transect lines, at a minimum distance of 450 m from other previously established plots, to carry out the vegetation surveys at various distances from water sources

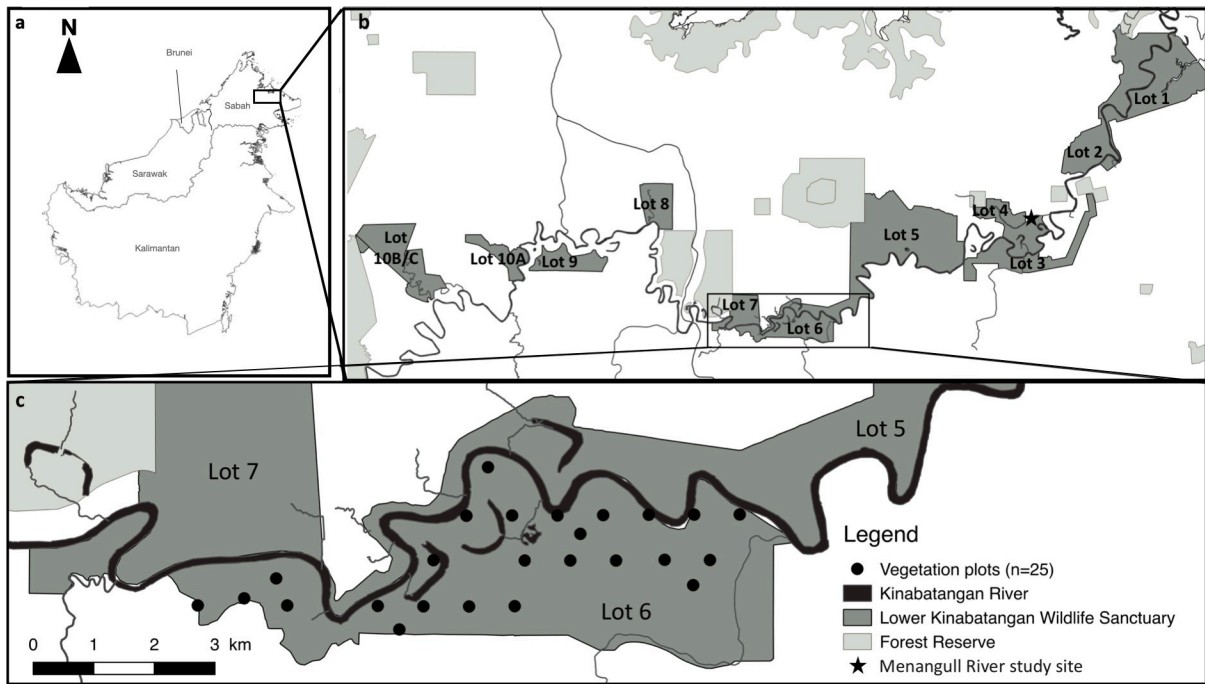

**Fig 1.** Map of the study area: a) The island of Borneo with the location of the Kinabatangan Floodplain in the State of Sabah; b) The Lower Kinabatangan Wildlife Sanctuary with its 10 protected lots (dark grey) and the Forest Reserves (light grey); c) Zoom on the study site with Lots 5, 6 and 7 and the location of the 25 vegetation plots inside Lot 6. Subpanel a) was generated using shapefiles from GADM (https://gadm.org), while subpanels b) and c) are based on the map provided in [38], licensed under Creative Commons Attribution 4.0.

(i.e., river, tributaries and oxbow lakes). In total, 10 plots were sampled < 250 m from water sources, and 15 were set up between 250 m and 1 km from water sources (Fig 1). In each plot, trees (Diameter at Breast Height, DBH ≥ 10cm) and vines (DBH ≥ 5cm) were measured and identified to the species level. When identification on site was not possible, leaf samples were collected and sent to the Forest Research Centre (Sandakan) to be identified by botanical experts. A leaf sample of each plant species was collected to build the local DNA reference database (see below "Building plant DNA databases").

## Direct behavioural observations

In riparian forests, proboscis monkeys regularly return to the forest edge along rivers, where they sleep at night [51 but see 30]. Therefore, in the late afternoon (04.30 pm) we conducted boat-based surveys to search for proboscis monkeys settling at their sleeping sites along a pre-established 21 km section of the Kinabatangan River. Over the course of a single month, we aimed to find proboscis monkey groups along different parts of the river to cover the whole study site and avoid observing the same groups repeatedly. We selected the first proboscis monkey group that we found and conducted behavioural observations from 17.00 h to 18.00–18.15 h. During the pilot fieldwork (2015), we recorded feeding observations by using *ad libitum* sampling [52]. In 2016 (January to June, November and December) and 2017 (January to March), we used the instantaneous scan sampling method [52], and recorded each individual behaviour (resting, feeding, moving, other [see 42]) at a 10-min interval. A 'feeding occurrence' was defined as a record of a scanned individual observed eating. Between scan intervals, we used the *ad libitum* sampling method to record potentially shorter and rarer feeding events that may occur between two consecutive scans. The following morning (from 6.00 h to 7.00 h), we conducted additional behavioural observations on the same group. After the group left the riverside, we identified the plant species (tree, vine, herbaceous plant) the individuals had been observed feeding on. When we were not immediately able to identify the plant to the species level, we collected samples of the leaves and sent them to the Forest Research Centre (Sandakan, Sabah) to be identified by botanical experts. A leaf sample of each feeding species was collected to build the local DNA reference database (see below "Building plant DNA databases"). Access to the fieldwork and material collection received approval from the Sabah Biodiversity Council (access licence number: JKM/MBS.1000-2/2 JLD.3 (67)). Plant and faeces samples were exported for laboratory procedures in Belgium under the export licence (licence number: JKM/MS.1000-2/3 JLD.2).

## DNA metabarcoding method

In this study, the metabarcoding approach was used to compare DNA sequences (trnL (UAA)) retrieved in proboscis monkey faeces to two DNA references databases: global and local. Indeed, the concomitant use of global and local databases has been shown to improve taxonomic resolution [see 15].

**Building plant DNA databases.**   The global "EMBL database" was built by extracting 16,711 trnL sequences from the European Molecular Biology Laboratory (EMBL) library (release 129) using the ecoPCR program [53]. To build the local "LKWS database", we sampled 475 plants (n = 242 species) in the study site. These plants were available in botanical plots (i.e., our 25 botanical plots, phenology plots [30], etc.), observed to be consumed by proboscis monkeys at the riverside, or randomly encountered in the study site (e.g., during forest walks). Voucher specimens were collected for each plant and deposited at the Herbarium of the Université Libre de Bruxelles (BRLU). In addition, for each sample, a leaf was stored in silica gel until DNA extractions. NucleoSpin 96® Plant II Kit (Macherey-Nagel) was used to extract

total plant DNA from 20mg of dried plant material, following the manufacturer's manual. DNA extracts were amplified using universal c-d primers (c: 5'- CGAAATCGGTAGACGCTACG–3'; d: 5'- GGGGATAGAGGGACTTGAAC-3'), amplifying the whole chloroplast trnL (*UAA*) intron [54]. Sequencing reactions were performed both in forward and reverse using Big Dye v.3.1 chemistry (Applied Biosystems). The products were sequenced on a 3730 DNA Analyzer (Applied Biosystems). Finally, for each plant, a shorter fragment of the trnL intron (the P6 loop sequence, 10–143 bp [25]) was extracted and used to build the local reference database. Sequence alignments, editing, and assembling were performed in CodonCode Aligner software (version 7.0.1, CodonCode Corporation). Some sequences could not be included in the local "LKWS database" as their amplification or sequencing had failed (e.g., *Garcinia parvifolia*, *Nauclea subdita*, etc.). In total, the local "LKWS database" includes 205 plant species, belonging to 146 genera and 61 families, and is available on GenBank (accession numbers from OR670713 to OR670948).

**Faecal sampling.** In the morning, we searched for proboscis monkey faeces under their sleeping trees. Proboscis monkey faeces were easily identifiable, generally properly shaped and soft and spread out on leaves or on the ground. When sympatric diurnal primates spent the night in or neighbouring (within 10 m) proboscis monkeys' sleeping trees, we did not collect faecal samples. Two faecal samples were collected per group and stored in empty tubes. Back at the research station, one sample was manually analysed to look for intact seeds [see in 55], the other was kept for DNA analysis and stored using the two-step method [56]: in 95% ethanol for 24 hours followed by silica desiccation. Indicating silica gel was replaced every day until the faecal sample was dried. During the study, 155 faecal samples were collected for DNA metabarcoding analyses (mean ± SD = 10 ± 4 faeces month$^{-1}$, range = 1–12 faeces month$^{-1}$).

**DNA extraction from faecal samples.** Total DNA was extracted from 25 mg of faecal samples (unsorted faeces, such as recommended in [19]), using the DNeasy® Blood and Tissue Kit (Qiagen), and following the manufacturer's instructions. Concentration of DNA extracts was assessed using Qubit® dsDNA HS Assay Kit (Invitrogen).

**Library preparation and sequencing.** Library preparation was conducted following the protocol "Illumina's 16S Metagenomic Sequencing Library Preparation file", but using PCR profiles described in [57] and g-h primers with adapters, designed by [25], to amplify plant chloroplast trnL (*UAA*). The Illumina workflow consisted in two consecutive PCR, with first PCR used to amplify P6 loop amplicon and second PCR for dual indexing, using Nextera XT Index Kit (Illumina). Two PCR replicates were performed for all faecal samples, reaching a total of 310 samples (n = 155 faecal samples and 155 PCR replicates).

Profiles of the different PCR products were checked using the QIAxcel (Qiagen), following the manufacturer's instructions. Considering these DNA concentrations, we finally prepared a mix containing approximately similar PCR product quantities of each faecal sample. Paired-end Illumina Sequencing (MiSeq) was conducted at the GIGA-Genomics platform (Université de Liège) using the MiSeq Reagent Kit v2 (2 x 150 bp).

**Sequence analyses: Taxon assignation and filtering steps.** Illumina sequences were demultiplexed from each other according to their tags (5xx and 7xx). Paired-end reads from faecal DNA were analysed using OBITools and a set of associated programs [58]. Paired-end reads were aligned and merged using the illuminapairedend program. Primers were identified and removed using the ngsfilter program. We only considered sequences with a maximum of two errors on g-h primer sequences. Identical sequences were clustered together, using the obiuniq program, saving the information about their occurrences among samples. Sequences shorter than 10 bp, longer than 150 bp, or with an occurrence lower or equal to 10 were deleted, using the obigrep program. Then, we assigned a status "Head", "Singleton", or

"Internal" to each sequence, and removed all the "Internal" sequences (potential PCR chimeras) using the obiclean program, as described in [15].

The EcoTag program [17] was used for taxon assignation of each sequence, using both the global "EMBL database" and the local "LKWS database". This program compared sequences from faeces to sequences in both databases and provided similarity scores. The EcoTag program is based on an algorithm that provides a unique taxon (with the best match, i.e., the highest score) to each unique sequence. This unique taxon corresponds to the last common ancestor node in the NCBI taxonomic tree of all the taxids that best matched against the query sequence [15]. For each taxon assignation, we selected the database (local or global) providing the highest best match, choosing preferentially the local database when both databases provided identical assignation scores. When the assignation was conducted with the local database, the sequence was assigned to the species level when the score = 1, to the genus level when the score was between 0.98–1, and to the family level when the score was between 0.95–0.98. When the global database was used for assignation, sequences with scores between 0.95 and 1 were only assigned to the family level (i.e., we never assigned to species or genus level based on global database). Using both databases, sequences with a score < 0.95 were not assigned and were discarded from further analyses.

To exclude the sequences that may result from PCR artefacts or contamination, we applied three filtering steps. Firstly, among the 681 unique sequences or MOTUs (Molecular Operational Taxonomic Units), we only retained sequences (n = 421) occurring in both PCR replicates. Secondly, we only kept sequences that could be assigned to the family level (n = 276), removing all sequences with an assignation score < 0.95 (n = 125), as well as a few sequences that were only assigned to subclass, order, or "no rank" level (n = 20). Finally, to avoid including sequences from species that do not constitute the ordinary diet of proboscis monkey but could result from external contamination (e.g. pollen deposited on faeces) or correspond to rarely ingested plants, we removed all sequences that met the two following conditions: 1) occurrence in < 10% of faecal samples and 2) account for < 1% of the mean read coverage of samples in which these sequences were found (light grey dots in Fig 2). This final filtering step reduced the number of MOTUs that we kept for further diet analyses to 100 (black dots in Fig 2). This filtering approach is conservative, as 85.3% of unique sequences were eliminated but the large majority of all reads found belonged to one of the 100 MOTUs (c. 82%, see results).

## Statistical analyses

Analyses were conducted with R (version 3.4.3) [59]. Species accumulation curves were used to assess the efficiency of our plant and faecal sampling efforts: we analysed the cumulative number of plant species against the number of botanical plots, as well as the cumulative number of MOTUs against the number of sampled faeces. Species accumulation curves were generated with the *vegan* package [60], using the *specaccum* function. Total richness was estimated using the Chao estimator from the *poolaccum* function. Spearman rank correlations were carried out to test the relation between the number of reads per faecal sample and the richness of detected MOTUs, as well as the relation between the frequency of occurrence ($F_O$) of plant families in the habitat (plot survey) and the $F_O$ of plant families recorded in faeces, by DNA metabarcoding analyses. To assess seasonal variations, we grouped the same months of different years together (i.e., May referring to May 2015 and May 2016, March to March 2016 and March 2017, etc.). Analyses of variances (Anova or Kruskal-Wallis, according to the distribution of model residuals and the homogeneity of variances) were conducted using the *userfriendlyscience* package [61], to investigate whether the number of MOTUs detected per faecal sample and the total richness of MOTUs varied between study months. To assess whether

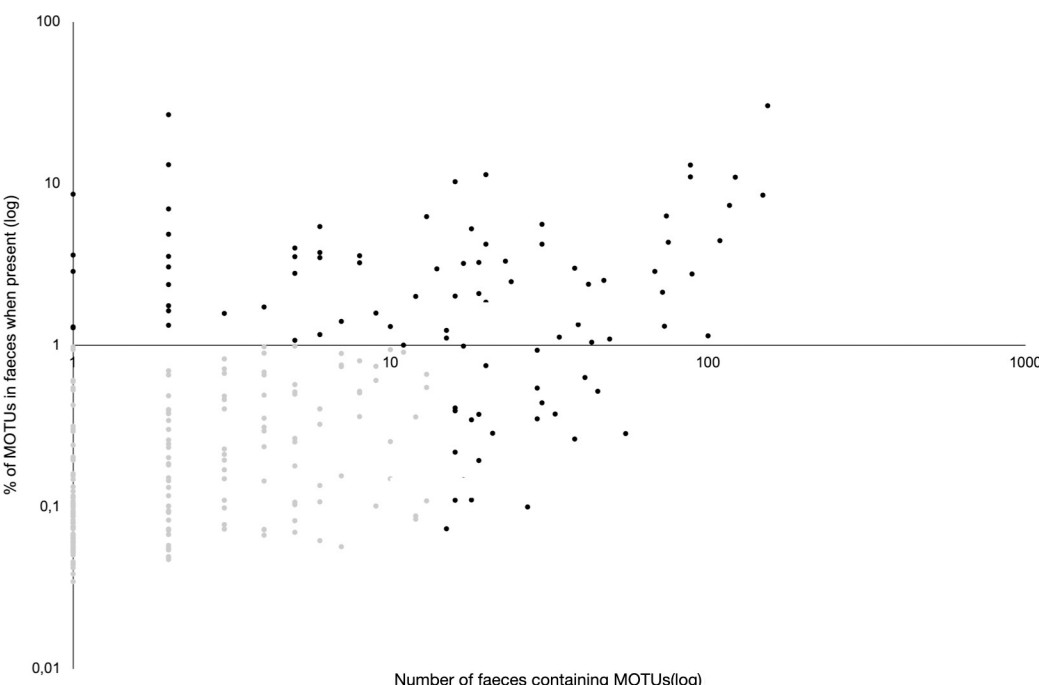

**Fig 2. Distribution of the 276 MOTUs according to their occurrence in faeces and their mean proportion of faecal read coverage in samples where they were detected.**

MOTUs showed a seasonal trend, $\chi^2$ tests were carried out, on Excel version 16.24 (2019), comparing the Frequency of occurrence ($F_O$) of MOTUs between wet and dry seasons. This analysis only took the abundant MOTUs (n = 65) occurring in $\geq$ 10% of faeces into account. We also tested if there was a correlation between the presence-absence (PA) of *Nauclea* spp. seeds in faeces (manual analyses, see details in [55]), and the PA of corresponding MOTUs (detected by DNA metabarcoding). Although these two analytical methods were not conducted on identical faecal samples, we compared faeces that belonged to the same proboscis monkey group, and that we collected on the same day, at the same location. By combining feeding data collected by both methods, we generated a list of plant taxa illustrating the overall diet of proboscis monkeys. To assess the congruence and complementarity of the two methods, we plotted Venn diagrams with plant families, genera and species or MOTUs, using the *draw.pairwise.venn* function from the *VennDiagramm* package [62].

## Results

### Vegetation survey

The vegetation survey covered 6.25 ha (25 plots of 2,500 m$^2$), where 4,457 plants were recorded, including 3,342 trees and 1,115 vines. We recorded 201 plant taxa belonging to at least 174 species, 116 genera and 51 families. The five most abundant plant families were Malvaceae, Dilleniaceae, Euphorbiaceae, Phyllanthaceae, and Lophopyxidaceae, accounting for 55% of all plants. The five most abundant tree species, accounting for 42.6% of all trees, were *Dillenia excelsa* (14.7%), *Mallotus muticus* (10.8%), *Colona serratifolia* (7%), *Antidesma puncticulatum* (5.6%) and *Vitex pinnata* (4.6%). The five most abundant vines, accounting for 70.5% of all vines, were *Lophopyxis maingayi* (41.1%), *Entada rheedii* (10.1%), *Bridelia stipularis* (8.3%), *Dalbergia stipulacea* (7.6%) and *Croton triqueter* (3.4%). The cumulative number of

plant species sampled in the 25 botanical plots did not reach an asymptote, showing that species diversity is high in our study site (S1 Fig).

## Direct behavioural observations

During the study (instantaneous scan sampling method: 2016–2017), we conducted 93 morning and 109 afternoon surveys along the Kinabatangan River. We carried out 7,391 individual observations: 2,220 in the morning and 5,171 in the afternoon. We recorded a total of 1,668 feeding occurrences, with proboscis monkeys consuming 42 plant taxa, belonging to 23 families, 32 genera and 42 species (S1 Table). Leaves accounted for 72% of proboscis monkey feeding occurrences, followed by fruits and flowers (5% and 0.8%, respectively); 22% of feeding items could not be categorized, as we conducted our observations from boat and could not always distinguish the plant part ingested (e.g., the individual shows its back). A seasonal trend was observed in fruit feeding behaviour at the riverside, with proboscis monkeys consuming fruits significantly more often during the wet season (8%) than the dry season (1%) ($X^2 = 14.6$, df = 1, p<0.001). The five most consumed families by proboscis monkeys along the river were the Moraceae, Rubiaceae, Tetramelaceae, Malvaceae, and Lamiaceae. The most consumed species were *Ficus racemosa*, *Octomeles sumatrana*, *Nauclea orientalis*, *Pterospermum elongatum*, and *Vitex pinnata*; with *F. racemosa* accounting for 70% of proboscis monkey leaf-eating occurrences along the riverbanks. *Nauclea orientalis* constituted 88% of all fruit-feeding occurrences, despite fruit-eating being rarely observed at the riverside.

By adding *ad libitum* sampling observations between scan intervals and the additional data collected from May to August 2015, we recorded a total of 67 plant taxa that the proboscis monkeys were feeding on at the riverside, belonging to at least 30 families, 48 genera and 62 species (S1 Table). The behavioural observation methods provided a high resolution, allowing us to identify 81% and 16% of the consumed plants until species and genus levels respectively, with only two plants remaining unidentified.

## DNA metabarcoding method

The next-generation sequencing produced a total of 8,986,271 reads (mean ± SD = 28,988 ± 8,785 reads per sample (n = 310), range = 224–60,199), corresponding to 681 unique sequences or MOTUs (Molecular Operational Taxonomic Units). After applying the different filtering steps (see in Methods, Fig 2), 100 MOTUs (S2 Table) were retained with 7,299,005 reads, corresponding to 81.2% of the initial total count (mean ± SD = 46,639 ± 17,187 reads per faecal sample, range = 5,137–81,867). Each faecal sample contained on average 19 ± 8 MOTUs (range = 1–39), with no significant difference between months (Anova: F = 1.50, df = 9, p = 0.15). The total richness of MOTUs detected each month did not vary significantly during the study (Kruskal-Wallis: $X^2 = 14$, df = 14, p = 0.45), with an average of 60 (± 18 SD) MOTUs detected every month (range: 10–77, in June 2016 and July 2015, though only one faecal sample was collected in June 2016).

The mean (± SD) length of the 100 recorded MOTUs reached 49 (± 8) bp (range = 20–68 bp), with no correlation between the sequence length and its occurrence in faeces nor its count of reads (Spearman: S = 180,840; rho = -0.09; p = 0.4 and S = 181,620; rho = -0.09; p = 0.374, respectively). We found a significant correlation between the total number of reads per faecal sample and the MOTU richness (Spearman: S = 284,363; rho = 0.54; p<0.001). We also observed that the frequency of occurrences of families inside the habitat (25 plots) and the frequency of occurrences of families in faeces were significantly positively correlated (Spearman: S = 22,042; rho = 0.29; p<0.05).

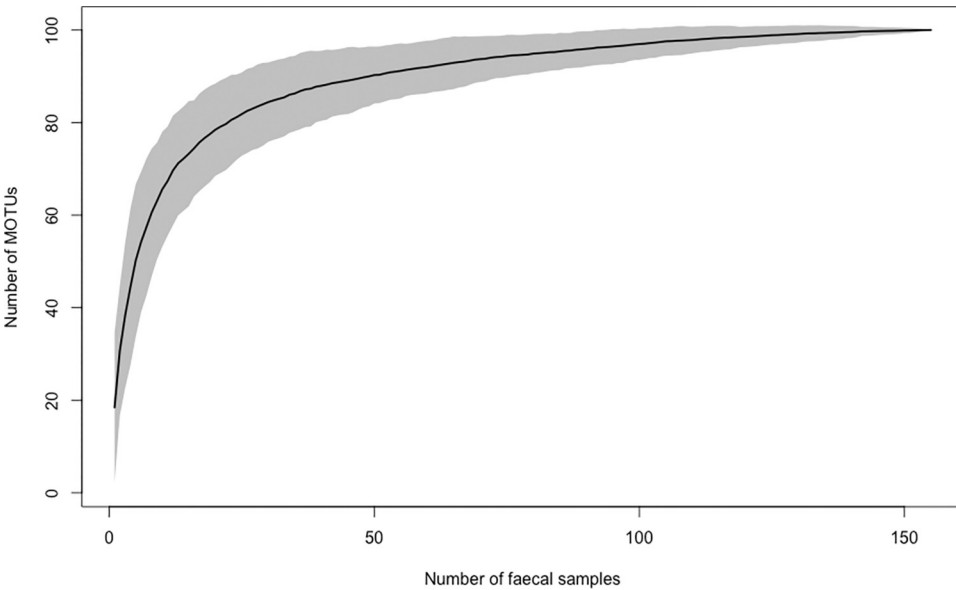

**Fig 3. Accumulation curve of plant DNA sequences (MOTUs) found in proboscis monkey faecal samples.** Faecal samples (n = 155) were collected in the Lower Kinabatangan Wildlife Sanctuary between May 2015 and March 2017.

The cumulative numbers of MOTUs approached an asymptote that was estimated at 106 MOTUs, suggesting that our sampling effort was sufficient to provide reliable information on proboscis monkey diet composition (Fig 3).

The diet of proboscis monkeys included at least 100 different plants, with 70% of MOTUs matching plant sequences available in the local "LKWS database". By using the two reference databases, 49, 18 and 33% of the 100 MOTUs were identified to the family, genus and species levels, respectively (Fig 4). Plant sequences were assigned to 39 families, 46 genera and 33 species. The top five plants recorded in proboscis monkey faeces belonged to *Bridelia*, *Ficus*, *Octomeles*, *Mallotus* and *Dracontomelon* genera. With a frequency of occurrences of 99%, *Bridelia* 1 (Phyllanthaceae) was the most detected plant taxa in this study (S2 Table). However, we could not assign this sequence to a unique species because the trnL P6 loop was not discriminant within the *Bridelia* genus. The most frequent families were the Phyllanthaceae, Moraceae, Leguminoseae, Euphorbiaceae and Tetramelaceae. The Leguminosae family is represented by the highest number of plant sequences (n = 14 MOTUs) (S3 Table), though only four of them could be identified to species level and the other 10 sequences were only identified until the family level.

The average frequency of occurrences ($F_O$, = the proportion of faeces containing the MOTU) was 19.5% (± SD = 21.3), indicating that on average most plant taxa were found in a fifth of the faecal samples.

**Seasonal variation.**   High numbers of plant taxa were recorded during both seasons, with 89 MOTUs detected from April to August (n = 64 faecal samples) and 98 from November to March (n = 91 faecal samples). We observed that $F_O$ of MOTUs varied significantly between seasons ($X^2$ = 140.31, df = 64, p<0.001). Seven of the MOTUs contributed the most to the seasonal trend ($X^2$ = 71.52, df = 6, p<0.001), with *Poikilospermum suaveolens*, *Lagerstroemia speciosa* and *Cayratia trifolia* more consumed during the wet season, and *Entada rheedii*, *Neolamarckia cadamba*, *Dillenia* 1 and *Pterospermum* 1 during the dry season (Fig 5).

**Correspondence with manual seed analyses.**   We did not find any correlation between the presence-absence (PA) of *Nauclea* spp. seeds in proboscis monkey faeces and the PA of Rubiaceae MOTUs in faeces (collected from the same group, the same day and at the identical

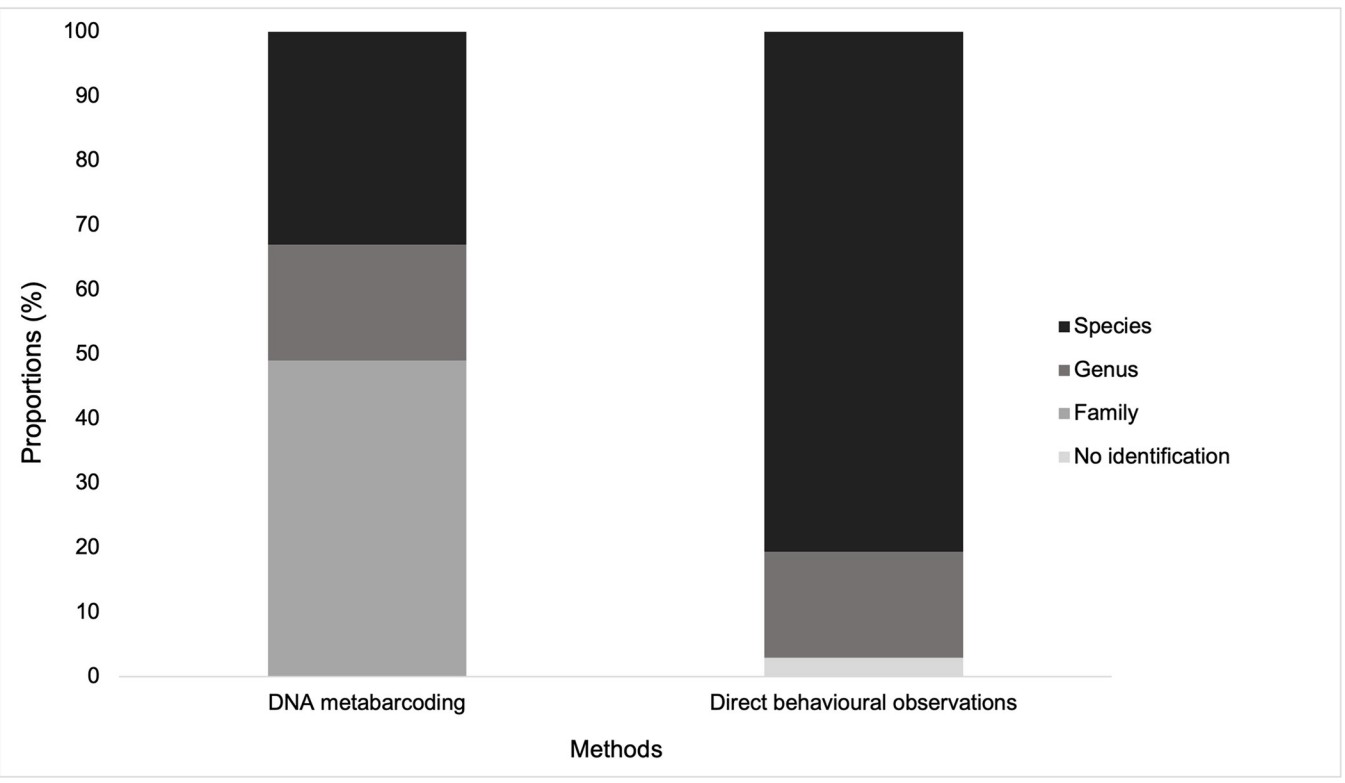

**Fig 4. Taxonomic resolution of the diet of proboscis monkeys using two methods: Direct behavioural observations and DNA metabarcoding.**

location) (i.e., trnL was not discriminant among species of the Rubiaceae family) (Spearman: S = 426,040; rho = -0.137; p = 0.118). When only taking the 75 faecal samples that were collected during months where 100% of manually analysed faeces contained *Nauclea* spp. seeds (n = 7 months: January to March 2016, and November 2016 to February 2017 [50]) into account, only 42 samples contained one (or more) of the four Rubiaceae MOTUs (potentially corresponding to *Nauclea* genus).

## Comparison of methods

Behavioural direct observations showed a higher resolution than DNA metabarcoding methods, as they provided identification to the species level for 81% of consumed plants in comparison to 33% with the DNA metabarcoding method (Fig 4).

The 20 top-key plants recorded by both methods shared three species (*Octomeles sumatrana*, *Dracontomelon dao* and *Cayratia trifolia*) and five genera (*Ficus*, *Mallotus*, *Vitex*, *Dillenia* and *Pterospermum*). Seven genera of the 20 top-key plants identified by direct behavioural observations were not recorded by DNA metabarcoding. However, three such genera (*Nauclea*, *Ludekia* and *Mitragyna*) belong to the Rubiaceae family, where trnL is not discriminant at the genus level, and two other genera (*Garcinia* and *Cleistanthus*) were not available in local and global reference databases. Therefore, only two genera (*Colona* and *Drypetes*) of the 20 top-key plants detected by direct behavioural observations were not recorded by the second method, although available in local and global reference databases, respectively.

Three vine species are listed in the 20 top-key plants recorded by behavioural direct observations, while at least six (potentially seven if considering *Bridelia* 1 to belong to the vine *Bridelia stipularis*, see discussion) are recorded by DNA metabarcoding (Table 1).

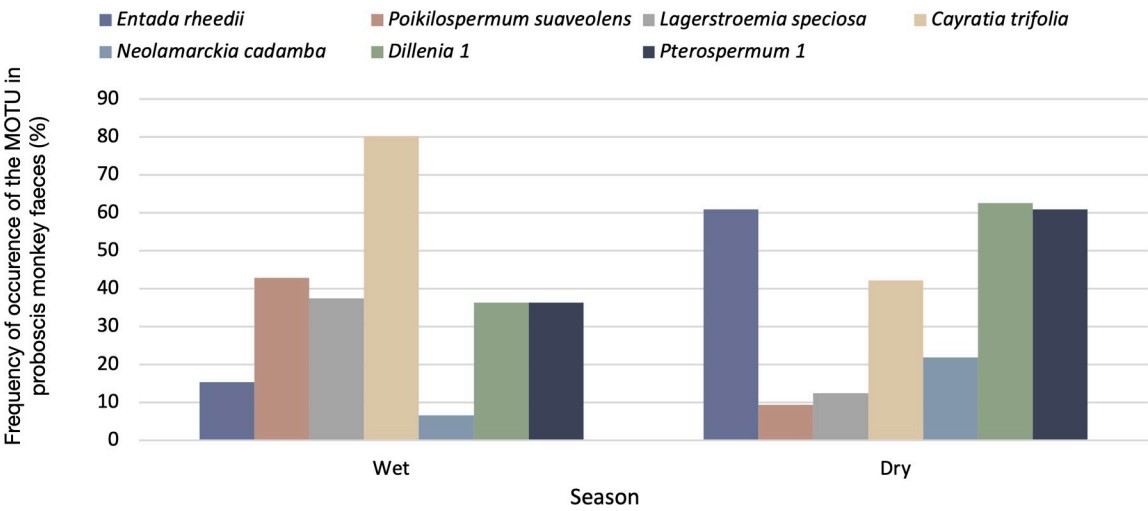

**Fig 5. Seasonal variation in the frequencies of occurrences of seven MOTUs detected in proboscis monkey faeces.**

## Combining methods: Overall proboscis monkey diet

To combine results from both methods (DNA metabarcoding and direct behavioural observations), we first listed all plants that were identified to the species level. Then, we added plants belonging to a new genus that had only been identified to the genus level. Finally, we added plants belonging to a new family that were only identified to the family level. Therefore, by combining both methods, we obtained a total of 89 plant taxa consumed by proboscis monkeys in our study site, belonging to 76 genera (including six unidentified genera) and 45 families (see S4 Table).

We observed that the DNA metabarcoding and direct behavioural observation methods provided congruent, but complementary, results: 24 families, 24 genera and 15 species were identified by both methods. At the family level, the congruence of methods was high with 80% of the families and 50% of the genera identified by direct behavioural observations that were also recorded by DNA metabarcoding. In fact, the DNA metabarcoding method provided 29 new plant taxa (including 18 identified species), belonging to 28 genera and 15 families, in addition to the first method (Fig 6).

Among the 18 new species identified as part of proboscis monkey diet, most of them were frequently detected during the study period (i.e., during at least eight months of the study, in a range of 14–61 faecal samples), except for three species, *Antirhea inaequalis*, *Dimocarpus* sp.1, *Crateva religiosa*, which were rarely found (n ≤ 2 samples, Table 2).

## Discussion

Our study reports the diet composition of multiple proboscis monkey groups inhabiting the riverine forests of the LKWS in Sabah, using two analytical methods: direct behavioural observations and DNA metabarcoding.

### Direct behavioural observations

Of the 67 plant species recorded by direct behavioural observations along the riverside, *Ficus racemosa* was the most consumed species (accounting for 50.8 % of feeding occurrences). It contrasts with previous studies conducted downriver, where *F. racemosa* was consumed less frequently [42, 44]. Nutritional content (i.e. protein), as well as plant abundance, are known to influence young leaf selection in proboscis monkeys [63, 64]. Variations in nutritional content

**Table 1. List of the 20 top-key plants recorded by behavioural direct observations and DNA metabarcoding methods.**

| Rank | Behavioural direct observations | | | DNA metabarcoding | | |
|---|---|---|---|---|---|---|
| | Species | Nb feeding occurrences | Fo | MOTUs | Nb occurrences in faeces | Fo |
| 1 | *Ficus* racemosa | 848 | 0.508 | *Bridelia* 1 | 154 | 0.994 |
| 2 | *Octomeles sumatrana* | 195 | 0.117 | *Ficus* 1 | 149 | 0.961 |
| 3 | *Nauclea orientalis* | 178 | 0.107 | *Octomeles sumatrana* | 122 | 0.787 |
| 4 | *Pterospermum elongatum* | 104 | 0.062 | *Mallotus* 1 | 117 | 0.755 |
| 5 | *Vitex pinnata* | 39 | 0.023 | *Dracontomelon dao* | 109 | 0.703 |
| 6 | *Ludekia borneensis* | 30 | 0.018 | *Cayratia trifolia* | 100 | 0.645 |
| 7 | *Mallotus muticus* | 28 | 0.017 | *Syzygium* 1 | 89 | 0.574 |
| 8 | *Colona serratifolia* | 20 | 0.012 | *Lophopyxis maingayi* | 88 | 0.568 |
| 9 | *Dracontomelon dao* | 16 | 0.010 | *Rubiaceae* 1 | 88 | 0.568 |
| 10 | *Cleistanthus sp.* | 15 | 0.009 | *Leguminosae* 1 | 75 | 0.484 |
| 11 | *Unknown sp.2* | 12 | 0.007 | *Vitex* 1 | 74 | 0.477 |
| 12 | *Ficus crassiramea* | 11 | 0.007 | *Dillenia* 1 | 73 | 0.471 |
| 13 | *Cayratia trifolia* | 11 | 0.007 | *Pterospermum* 1 | 72 | 0.465 |
| 14 | *Dillenia excelsa* | 10 | 0.006 | *Polyalthia* 1 | 68 | 0.439 |
| 15 | *Drypetes sp.* | 9 | 0.005 | *Caesalpinia sp.1* | 64 | 0.413 |
| 16 | *Mikania cordata* | 9 | 0.005 | *Derris elegans* | 61 | 0.394 |
| 17 | *Mallotus floribundus* | 6 | 0.004 | *Erycibe grandifolia* | 55 | 0.355 |
| 18 | *Mitragyna speciosa* | 6 | 0.004 | *Apocynaceae* 1 | 55 | 0.355 |
| 19 | *Garcinia parvifolia* | 6 | 0.004 | *Entada rheedii* | 53 | 0.342 |
| 20 | *Unknown sp.1* | 5 | 0.003 | *Lauraceae* 1 | 49 | 0.316 |

[i] Species and genera found in both methods are denoted in bold.

and species abundance between the downriver site from the previous studies [42, 44] and ours may explain these dietary differences. Indeed, *F. racemosa* is a common tree species along the riverbanks of the Kinabatangan River [65] and its high-quality young leaves, with high crude ash [66], may help explain why it was highly consumed during our observation sessions restricted at the riverside. Moreover, both previous studies collected feeding data during the entire day further from water sources [42, 44], where *F. racemosa* might occur less frequently.

Using direct behavioural observations, a particularly low level of frugivory–only 5% of feeding occurrences at the riverside–was recorded, in comparison to previous studies where fruits accounted for 26% [42], 40% [34] or even 50% [33] of proboscis monkey diet. Even among such few frugivory events, more fruit eating was reported during the wet season. This result needs to be taken cautiously, however, as fewer direct behavioural observations were conducted during the dry season (17% of total feeding occurrences) and a bias may thus exist because of the differences in sampling efforts between the two seasons. The lower fruit-feeding activity in the current study could, in part, be explained by the fact that our observation sessions were restricted spatially, to riverbanks, and temporally, to early mornings and late afternoons. Increased fruit-feeding activities might occur inland during the day and therefore be missed during boat-based surveys. Moreover, proboscis monkeys may preferentially feed on leaves at the riverside, at least in the late afternoon as sleep hours can then be devoted to digesting the fibrous materials [67]. Finally, young leaves were highly available throughout the study period in comparison to fruits and flowers (phenology survey data in [66]). Although Bennett and Sebastian [33] reported more fruit-feeding during boat-based surveys than we did in the current study, their results must also be taken cautiously as they recorded very few total feeding events (n = 34, as opposed to n = 1,668 in the current study). Finally, we found strong

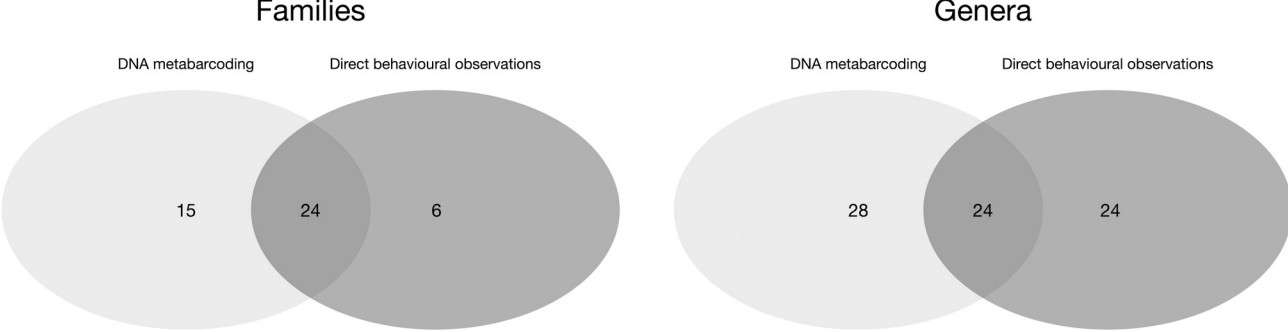

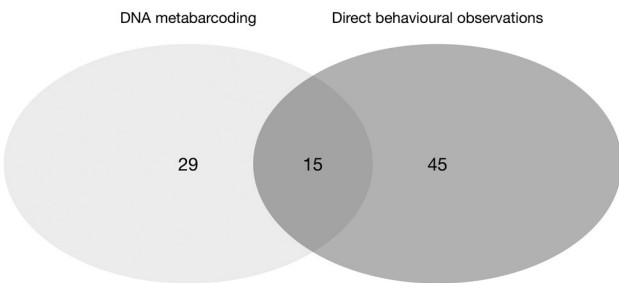

**Fig 6. Venn diagrams of plant families, genera and species detected by DNA metabarcoding and direct behavioural observation methods.**

**Table 2. List of the 18 plant species only recorded by the DNA metabarcoding method.**

| Family | Species | Number of faecal samples |
|---|---|---|
| Burseraceae | *Canarium denticulatum* | 38 |
| Calophyllaceae | *Mesua oblongifolia* | 14 |
| Capparaceae | *Crateva religiosa* | 1 |
| Convolvulaceae | *Erycibe grandifolia* | 55 |
| | *Merremia umbellata* | 23 |
| Cornaceae | *Alangium javanicum* | 15 |
| Lamiaceae | *Teijsmanniodendron bogoriense* | 16 |
| Lauraceae | *Cryptocarya ferrea* | 39 |
| Lecythidaceae | *Barringtonia pterita* | 19 |
| Leguminosae | *Derris elegans* | 61 |
| | *Entada rheedii* | 53 |
| Malvaceae | *Microcos crassifolia* | 29 |
| Phyllanthaceae | *Margaritaria indica* | 20 |
| Rubiaceae | *Neolamarckia cadamba* | 20 |
| | *Antirhea inaequalis* | 2 |
| Sapindaceae | *Dimocarpus longan* | 19 |
| | *Dimocarpus* sp.1 | 2 |
| Vitaceae | *Tetrastigma lanceolarium* | 41 |

evidence of fruit-feeding behaviour in proboscis monkeys in a concurrent study, with 77% of proboscis monkey faeces analysed in the study site containing intact seeds, with 98% of the seeds belonging to *Nauclea* spp. [55]. The high percentage of faeces containing seeds highlights the significance of fruits in proboscis monkey diet and supports the idea that fruit-feeding may have been missed during our observations at the riverside.

## DNA metabarcoding method

Using DNA-based analyses with only about 3.9 g of faecal samples (155 * 25 mg) was sufficient to record 100 plant taxa that proboscis monkeys were feeding on in the LKWS. Each month, 10 to 77 plant taxa were recorded in proboscis monkey faeces. A similar range was reported in a previous study, where proboscis monkeys consumed 36 to 82 plant species each month [42].

*Bridelia* 1 is the most abundant MOTU in this study, which was recorded in 99% of faecal samples. Two species of *Bridelia* were observed during the vegetation survey: the vine *B. stipularis* and the tree *B. insulana*. As *B. stipularis* is more available (n = 92, basal area/ha = 648 cm$^2$/ha) than *B. insulana* (n = 3, basal area/ha = 66 cm$^2$/ha), we could expect *Bridelia* 1 to refer to *B. stipularis* and *Bridelia* 2 rather than to *B. insulana*. Moreover, proboscis monkeys were observed to consume *B. stipularis* during boat-based surveys, and *B. stipularis* seeds were recorded in several monkey faeces [55]. The trnL sequence of *Bridelia* 1 is short, with a length of 41 bp. Although previous studies have reported that short sequences are likely to be over-represented, with short sequences occurring in higher numbers than long sequences [18, 68], we did not observe any correlation between the length of MOTUs and their occurrences in faeces nor their counts of reads in the present study.

A positive correlation was reported, at the family level, between plant abundance in the habitat and occurrences in faeces, suggesting that proboscis monkeys feed on the most available plants in the study site. Abundance is known to influence plant selection in proboscis monkeys, with the monkeys choosing the most abundant plants of their preferred food species [63]. However, we could not test this selection at the genus and species level as almost half of the MOTUs were only identified to the family level. A seasonal trend was observed for nine MOTUs. These species/genera seem to mostly be selected for their leaves, except for *Dillenia excelsa* and *Pterospermum macrocarpum* (updated name *P. diversifolium*) which were highly consumed for their flowers [42]. Young leaves were highly available throughout the year in our study site, whereas flowers were more abundant in the dry season, and fruits during the wet season [66]. These changes in food (flower) availability throughout the study may explain the seasonal pattern observed for some of the seven MOTUs (i.e., *Dillenia* 1 and *Pterospermum* 1 were more eaten in the dry season, when flowers were abundant).

## Comparison of methods

In this study, a higher number of plant taxa was recorded by DNA metabarcoding approach (n = 100) than by direct behavioural observation method (n = 67), though this latter was only conducted during boat-based surveys in late afternoons and early mornings, thereby only characterizing feeding habits occurring at the riverside. Based on a previous study, where 35 plant taxa were identified as part of the diet for banded leaf monkeys (*Presbytis femolaris*), although only six faecal samples were used [14], we expected to record large numbers of plants in the 155 faeces of proboscis monkeys.

It is interesting to note that more vines were potentially detected by DNA metabarcoding than by direct behavioural observations. In fact, vine consumption was not always easy to detect during boat-based surveys as we were at a certain distance from the group, and therefore feeding bouts on vines might have been missed in the middle of feeding events on tree leaves.

Moreover, plant identification was not always possible, especially when feeding vines were high in the tree canopy and no leaves could be sampled for identification. Therefore, we suggest that DNA metabarcoding might be a more appropriate method to record vine species that usually are not easy to detect visually.

Taxonomic resolution was better by direct behavioural observations than DNA metabarcoding, with 81% and 33% of the taxa identified to the species level, respectively. Using DNA metabarcoding, taxonomic resolution to the genus level reached 51% (including 33 MOTUs identified until the species level), which is similar to previous studies, conducted in tropical environments, where genus-level identification reached 40 to 61% of the sequence identifications [14, 15, 19]. Taxonomic resolution could be improved using different primer pairs, such as the second internal transcribed spacer of nuclear ribosomal DNA (ITS2) (see [21]).

One of the main biases of DNA-based diet studies remains the impossibility to discern which plant part has been consumed by the animal [7]. The presence of Rubiaceae MOTUs (DNA metabarcoding analyses) was not correlated with the presence of *Nauclea* spp. seeds (manual analyses, see [50]) in faeces collected the same day and at the same site. Moreover, of the faecal samples collected during months when all the manually analysed faeces contained *Nauclea* spp. seeds (n = 75 samples in seven months [50]), only 56% of the samples analysed by DNA metabarcoding contained at least one of the Rubiaceae MOTUs. This weak correspondence between both methods indicates that plastid markers are not appropriate to record frugivory events, as previously reported by [24] (but see exceptions in [19]).

## Combining methods: Overall proboscis monkey diet

We obtained a diverse dietary profile of proboscis monkeys consisting of at least 89 different plant taxa, from 76 genera and 45 families. A good overlap is reported between the two methods: 80% of the families and half of the genera detected by direct behavioural observations were also identified by DNA metabarcoding, with 75% of the genera recorded in at least a fifth of the faecal samples. These genera included *Ficus*, *Octomeles*, *Pterospermumm* and *Vitex* genera, which also accounted for an abundant component of the feeding occurrences reported by direct behavioural observations at the riverside. In fact, during our boat-based survey, we were more likely to record feeding bouts involving frequently consumed plant species, which were therefore more likely to be detected in large numbers of faecal samples. Similar results were reported in a previous study comparing field data and DNA-based analyses (DNA metabarcoding and metagenomics) to describe the diet of *Presbytis femolaris* in Singapore [14]. Therefore, we suggest that DNA metabarcoding was a reliable method to unveil the diet of proboscis monkeys in this study, particularly in regard to the large number of faeces sampled (n = 155), the long duration of the study period (14 months), and the size of the study area (a 21-km river transect, where many proboscis monkey groups range). Moreover, with respect to the proboscis monkeys' long gut retention times, we consider that faecal samples provide information on the plants they consumed a few days before defecation (see [15]), potentially allowing us to retrieve feeding events of rare items. Among the 15 families only detected by DNA metabarcoding, two (Malpighiaceae and Symplocaceae) were not reported in our study site, but may have been missed during our vegetation survey. The Symplocaceae family is known to be present in the Kinabatangan floodplain [65], as well as to be part of proboscis monkey diet in Sukau [42]. The Malpighiaceae family is also known to be found on the island of Borneo [69]. Finally, among the 18 species that were only detected by DNA metabarcoding, nine had already been documented as part of proboscis monkey diet in riverine habitats [42].

The number of plant taxa (n = 89) consumed in our study is within the range reported by two previous long-term studies, listing 36 [44] and 188 [42] plant species being consumed by

proboscis monkeys downriver from our study site. When comparing the composition of the plant lists obtained in these three studies, we observed overlap and differences. We identified 37% and 43% of the genera identified by Matsuda and colleagues (n = 47 in [42]) and Boonratana (n = 10 in [44]), respectively. We also recorded 22 genera that had not yet been described as consumed by proboscis monkeys in the LKWS [42, 44]. Moreover, six of these new genera (*Octomeles*, *Ludekia*, *Colona*, *Mikania*, *Mitragyna* and *Polylthia*) are of main importance, belonging to the 20 top-key plants detected by direct behavioural observations or DNA metabarcoding. As 84% of feeding occurrences reported in [44] included unidentified species, we will further compare our diet data to the most comprehensive study [42] conducted along the Menanggul River (Lot 4 of the LKWS, Fig 1).

Several factors may explain the differences in proboscis monkey diet reported in our study and in Matsuda and colleagues [42]. Firstly, although both study sites are close from each other (approximately 30 km) and characterized by riverine forests where *Mallotus* and *Lophopyxis* are common, plant availability differs: while *Croton*, *Excoecaria* and *Eugenia* are abundant genera in the Menanggul riverine habitat, our study site is dominated by *Dillenia*, *Colona* and *Syzygium*. However, sampling methods used to carry out vegetation surveys differ between studies (trail transects in [42], botanic plots in the present study), resulting in plant availabilities that are hard to compare. Cumulative numbers of plant species sampled in both studies did not reach asymptotes, suggesting that the plant diversity is high and that further sampling efforts are required.

Secondly, methods used to record feeding data differed between studies: while Matsuda and his colleagues [42] conducted continuous focal follows from dawn to dusk, we carried out scan and *ad libitum* sampling restricted to the riverside in early mornings and late afternoons, and therefore not including the entire feeding behaviour of the monkeys. We also used DNA-based analyses to study the diet of proboscis monkeys, whereas Matsuda and colleagues [42] did not. However, as trnL P6 loop is an appropriate primer targeting chloroplast-rich tissues (i.e., leaves and stems), we suggest that plant species mostly consumed for their fruits or flowers may be detected less often. The number of MOTUs (n = 100) we obtained is likely a conservative estimate of proboscis monkey diet, as we used several filtering steps to remove potential erroneous sequences (i.e., PCR artefacts or contaminations), leading to an elimination of 85.3% of the unique sequences. These reasons could explain why the plant richness recorded in our study was lower than in [42].

## Conclusion

The present study enhances our knowledge on the feeding ecology of proboscis monkeys inhabiting riverine forests of the Kinabatangan floodplain. The two methods used in this study provided congruent and complementary results, both having their advantages and limitations. While DNA metabarcoding is a non-invasive tool providing reliable diet assessment for elusive species or non-habituated animals, the method is not very appropriate to infer quantitative information about the relative abundance of consumed plant taxa [18, 27], and the importance of species eaten for their fruits may be underrepresented. In addition, plants ingested rarely or accidentally may be overrepresented, although only considering plant taxa detected in multiple faecal samples should help overcome this bias [14]. The main advantage of direct behavioural observations remains the possibility to record which plant part is consumed by the individual, as well as to which specific tree or vine it belongs [14] (i.e., useful data for further nutritional composition analyses). However, following animals in the field may not always be an easy or a possible task (i.e., swamp or inundated forest, unhabituated animals).

To conclude, whether new technologies, such as DNA metabarcoding, allow to rapidly screen an area to unveil the diet of multiple species (see [20]), spending time in the field to conduct direct behavioural observations, even in spatially and temporally restricted conditions, such as in the present study, still provides valuable feeding data (i.e.; item consumption, better species resolution). Furthermore, species of the 20 top-key plants consumed by proboscis monkeys, such as *F. racemosa*, *O. sumatrana*, *N. orientalis*, *Pterospermum elongatum*, *Mallotus muticus*, *Dracontomelon dao* or *Cayratia trifolia* should be considered of high importance when developing future conservation strategies, such as reforestation programs and corridor establishment.

## Supporting information

**S1 Fig. Accumulation curve of plant species found in 25 botanical plots in Lot 6 of the LKWS.**
(DOCX)

**S1 Table. List of plant taxa (n = 67) consumed by proboscis monkeys at the riverside (recorded by instantaneous scan and *ad libitum* sampling observation methods, from May 2015 to March 2017).**
(DOCX)

**S2 Table. List of food plant taxa (n = 100) recorded in proboscis monkey faeces using DNA metabarcoding.**
(DOCX)

**S3 Table. List of food plant families (n = 39) recorded in proboscis monkey faeces.**
(DOCX)

**S4 Table. List of the plant taxa (n = 89) consumed by proboscis monkeys in the study site, combining both methods.**
(DOCX)

## Acknowledgments

We thank the Sabah Biodiversity Centre and the Sabah Wildlife Department for their permission and support to this research. We also thank the Forest Research Centre (FRC) members, especially Dr. Joan Pererra, for the identification of many plant samples. We gratefully thank all research assistants and students at Danau Girang Field Centre (DGFC) for their precious help in the field and Dr Senthilvel KSS Nathan for his contribution. Special thank you to Oriana Bhasin for assisting us in the early-morning faeces sampling. We extend our gratitude to Laurent Grumiau, Esra Kaymak and Jérémy Migliore for their assistance during the laboratory work, as well as Latifa Karim for technical support for the MiSeq sequencing (GIGA-Genomics platform).

## Author Contributions

**Conceptualization:** Valentine Thiry.

**Data curation:** Valentine Thiry, Arthur F. Boom.

**Formal analysis:** Valentine Thiry, Arthur F. Boom.

**Funding acquisition:** Valentine Thiry.

**Investigation:** Valentine Thiry, Arthur F. Boom, Danica J. Stark, Olivier J. Hardy, Benoit Goossens.

**Methodology:** Valentine Thiry, Arthur F. Boom, Danica J. Stark, Regine Vercauteren Drubbel, Benoit Goossens.

**Project administration:** Martine Vercauteren.

**Resources:** Olivier J. Hardy, Sylvia Alsisto.

**Supervision:** Danica J. Stark, Olivier J. Hardy, Roseline C. Beudels-Jamar, Regine Vercauteren Drubbel, Martine Vercauteren, Benoit Goossens.

**Visualization:** Valentine Thiry.

**Writing – original draft:** Valentine Thiry.

**Writing – review & editing:** Valentine Thiry, Arthur F. Boom, Danica J. Stark, Olivier J. Hardy, Roseline C. Beudels-Jamar, Regine Vercauteren Drubbel, Sylvia Alsisto, Martine Vercauteren, Benoit Goossens.

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
