## [Decision Letter · Decision Letter 0]

23 Aug 2024

PONE-D-24-28309Using DNA metabarcoding and direct behavioural observations to identify the diet of proboscis monkeys (Nasalis larvatus) in the Kinabatangan FloodplainPLOS ONE

Dear Dr. Boom,

Thank you for submitting your manuscript to PLOS ONE. After careful consideration, we feel that it has merit but does not fully meet PLOS ONE’s publication criteria as it currently stands. Therefore, we invite you to submit a revised version of the manuscript that addresses the  few minor points raised during the review process.

We look forward to receiving your revised manuscript.

Kind regards,

Wolfgang Blenau

Academic Editor

PLOS ONE

Journal requirements: 1. When submitting your revision, we need you to address these additional requirements. Please ensure that your manuscript meets PLOS ONE's style requirements, including those for file naming. The PLOS ONE style templates can be found at https://journals.plos.org/plosone/s/file?id=wjVg/PLOSOne_formatting_sample_main_body.pdf and https://journals.plos.org/plosone/s/file?id=ba62/PLOSOne_formatting_sample_title_authors_affiliations.pdf. 2. Thank you for stating the following financial disclosure:  [FNRS (Fonds de la Recherche Scientifique) Fonds Léopold III – pour l’Exploration et la Conservation de la Nature asbl. FNRS Gustave Boël-Sofina Fellowship].  Please state what role the funders took in the study.  If the funders had no role, please state: ""The funders had no role in study design, data collection and analysis, decision to publish, or preparation of the manuscript."" If this statement is not correct you must amend it as needed. Please include this amended Role of Funder statement in your cover letter; we will change the online submission form on your behalf. 3. Thank you for stating the following in the Acknowledgments Section of your manuscript: [We thank the Sabah Biodiversity Centre and the Sabah Wildlife Department for their permission and support to this research. We also thank the Forest Research Centre (FRC) members, especially Dr. Joan Pererra, for the identification of many plant samples. We thank all the financial supporters: the FNRS (Fonds de la Recherche Scientifique), the Fonds Léopold III – pour l’Exploration et la Conservation de la Nature asbl., and the FNRS Gustave Boël-Sofina Fellowship 2016. We gratefully thank all research assistants and students at Danau Girang Field Centre (DGFC) for their precious help in the field and Dr Senthilvel KSS Nathan for his contribution. Special thank you to Oriana Bhasin for assisting us in the early-morning faeces sampling. We extend our gratitude to Laurent Grumiau, Esra Kaymak and Jérémy Migliore for their assistance during the laboratory work, as well as Latifa Karim for technical support for the MiSeq sequencing (GIGA-Genomics platform).]We note that you have provided funding information that is not currently declared in your Funding Statement. However, funding information should not appear in the Acknowledgments section or other areas of your manuscript. We will only publish funding information present in the Funding Statement section of the online submission form. Please remove any funding-related text from the manuscript and let us know how you would like to update your Funding Statement. Currently, your Funding Statement reads as follows:   [FNRS (Fonds de la Recherche Scientifique) Fonds Léopold III – pour l’Exploration et la Conservation de la Nature asbl. FNRS Gustave Boël-Sofina Fellowship]. Please include your amended statements within your cover letter; we will change the online submission form on your behalf. 4. Please provide a complete Data Availability Statement in the submission form, ensuring you include all necessary access information or a reason for why you are unable to make your data freely accessible. If your research concerns only data provided within your submission, please write "All data are in the manuscript and/or supporting information files" as your Data Availability Statement. 5. Please include your full ethics statement in the ‘Methods’ section of your manuscript file. In your statement, please include the full name of the IRB or ethics committee who approved or waived your study, as well as whether or not you obtained informed written or verbal consent. If consent was waived for your study, please include this information in your statement as well. 

Reviewers' comments:

Reviewer's Responses to Questions

**Comments to the Author**

1. Is the manuscript technically sound, and do the data support the conclusions?

Reviewer #1: Yes

Reviewer #2: Yes

2. Has the statistical analysis been performed appropriately and rigorously? 

Reviewer #1: Yes

Reviewer #2: Yes

3. Have the authors made all data underlying the findings in their manuscript fully available?

Reviewer #1: No

Reviewer #2: Yes

4. Is the manuscript presented in an intelligible fashion and written in standard English?

Reviewer #1: Yes

Reviewer #2: Yes

5. Review Comments to the Author

Reviewer #1: Using DNA metabarcoding and direct behavioural observations to identify the diet of proboscis monkeys (Nasalis larvatus) in the Kinabatangan Floodplain

Thiry et al.

reviewed by M. Clauss Zurich (does not do anonymous reviews)

This is a very good manuscript - a pleasure to read, and proof of a thorough and critical approach. How the different mtethods are compared, and the weaknesses of each outlined, is very clear and insightful. Thank you.

I have only minor comments - some references seem to have been mixed up (see attached word file). E.g., the Nijboer thesis 2006 is cited - but it consists either of individual chapters that should be cited seaprately, or of published papers that should be cited. Some citation numbering seems to be confused.

I also have a very few suggestions for grammar or spelling corrections in there.

I recommend to add 'seed' in the subheading 'correspondence with manual analyses' - see attached.

There are statements that fruit feeding differs by season, and for this, p-values are given but not the magnitude of the measurement. Note I think this paper is brlliant so this criticism is smartassing: it is common fashion in biology to value a p-value higher than an indication of magnitude, but it is the latter that is important biologically, the p-value is just thr prerogative that we can talk about the magnitude. So many people forget the magnitudes ... so if this is 5% fruit feeding vs. 6% fruit feeding, even if significant - a completely different interpretation than if it is 2% compared to 22% ... Please add that. Real biologists care about actual magnitudes, not about significance per se :-)

in l. 57 NIRS is mentioned - I thought this was a method to estimate nutrient content, not botanical composition - but I might be wrong. If you cite something on botanical composition here, forget my comment. If you cite this for nutrient content, I recommend to delete the NIRS from the list.

When reading for the first time, I was sceptical about the first sentence of the abstract, but the way this is not hyped in the main manuscript makes it ok, I think.

I could not find the GenBank entries, but that might have been my mistake. I recommend that for the final version, the access is ensured.

Thank you for having me review such a nice and concise manuscript!

sincerely. mclauss

Reviewer #2: I have also added comments in the manuscript.

Add one statement to indicate problem statement why Proboscis monkey and why in LKWS

The written manuscript is very good especially to involve endangered species. I would suggest to add more examples of research been done in Malaysia, or Asia regarding the diet intake using metabarcoding approach.

One section in introduction sound like or supposed to be in material and method section.

Add date/period of barcoding date base is generated

Add details on fecal samples, group information, etc in a Table

It would be more interesting to add details on metabarcoding data based on group, age and gender.

Revise the conclusion section

6. PLOS authors have the option to publish the peer review history of their article (what does this mean?). If published, this will include your full peer review and any attached files.

Reviewer #1: **Yes: **Marcus Clauss

Reviewer #2: No

---

## [Author Response · Author response to Decision Letter 0]

12 Dec 2024

We have addressed the reviewers' comments in the document attached to this resubmission (see the file Responses to the Reviewers' Questions and Comments.docx)

---

## [Editor Report · Decision Letter 1]

16 Dec 2024

Using DNA metabarcoding and direct behavioural observations to identify the diet of proboscis monkeys (Nasalis larvatus) in the Kinabatangan Floodplain, Sabah

PONE-D-24-28309R1

Dear Dr. Boom,

We’re pleased to inform you that your manuscript has been judged scientifically suitable for publication and will be formally accepted for publication once it meets all outstanding technical requirements.

Kind regards,

Wolfgang Blenau

Academic Editor

PLOS ONE
---

## [Editor Report · Acceptance letter]

23 Dec 2024

PONE-D-24-28309R1 

PLOS ONE

Dear Dr. Boom, 

I'm pleased to inform you that your manuscript has been deemed suitable for publication in PLOS ONE. Congratulations! Your manuscript is now being handed over to our production team.

Kind regards, 

on behalf of

Dr. Wolfgang Blenau 

Academic Editor

PLOS ONE